# Synthesis and Characterization of Silicone Contact Lenses Based on TRIS-DMA-NVP-HEMA Hydrogels

**DOI:** 10.3390/polym11060944

**Published:** 2019-05-31

**Authors:** Nguyen-Phuong-Dung Tran, Ming-Chien Yang

**Affiliations:** Department of Materials Science and Engineering, National Taiwan University of Science and Technology, Taipei 10607, Taiwan; thaonguyeng89@gmail.com

**Keywords:** TRIS, DMA, NVP, HEMA, silicone hydrogel, contact lens

## Abstract

In this study, silicone-based hydrogel contact lenses were prepared by the polymerization of 3-(methacryloyloxy)propyltris(trimethylsiloxy)silane (TRIS), N,N-dimethylacrylamide (DMA), 1-vinyl-2-pyrrolidinone (NVP), and 2-hydroxyethylmethacrylate (HEMA). The properties of silicone hydrogel lenses were analyzed based on the methods such as equilibrium water content, oxygen permeability, optical transparency, contact angle, mechanical test, protein adsorption, and cell toxicity. The results showed that the TRIS content in all formulations increased the oxygen permeability and decreased the equilibrium water content, while both DMA and NVP contributed the hydrophilicity of the hydrogels. The maximum value of oxygen permeability was 74.9 barrers, corresponding to an equilibrium water content of 44.5% as well as a contact angle of 82°. Moreover, L929 fibroblasts grew on all these hydrogels, suggesting non-cytotoxicity. In general, the silicone hydrogels in this work exhibited good oxygen permeability, stiffness, and optical transparency as well as anti-protein adsorption. Hence, these silicone hydrogel polymers would be feasible for making contact lens.

## 1. Introduction

Hydrogel is frequently used in fields including biology, medicine, material science and engineering. In particular, hydrogel has been applied in contact lenses due to impressive function such as wettability, biocompatibility, optical characteristic, and mechanical strength [1]. Nevertheless, hydrogel lens limits gas permeability, especially oxygen permeability. High oxygen permeability is very significant to maintaining the wearing comfort and to restrict hypoxia-related complications, such as dry eyes and corneal edema [2,3], as hypoxia frequently occurs in patients who wear contact lenses for long durations.

To reduce the shortage of oxygen supply, the hydrophobic materials containing siloxane groups such as hydroxyl-terminated polydimethylsiloxane (PDMS) and tris-(trimethyl-silyl-propyl-methacrylate) (TRIS) were used to improve the oxygen absorptivity of contact lenses. The cooperation of 2-methacryloyloxyethyl phosphorylcholine and vinyl ether terminated polydimethylsiloxane macro-monomer, improved the oxygen permeability of hydrogel contact lenses [4]. In addition, the polymerization of 2-hydroxyethyl methacrylate (HEMA) with PDMS provided more oxygen permeability than non-silicone hydrogel [5]. Moreover, a silicone hydrogel lens synthesized by the incorporation of oligosiloxane, N-vinyl-pyrrolidone (NVP), and N,N-dimethyl-acrylamide (DMA) also improved the oxygen uptake of hydrogel lens [6].

TRIS is a hydrophobic monomer with siloxane groups that can increase the oxygen absorbability of a conventional hydrogel lens. Because of the short chain length, TRIS is softer and lacks strength compared to PDMS [7]. Hence, TRIS-based hydrogel lenses exhibit lower mechanical strength and higher flexibility than PDMS-based hydrogel lenses. Although TRIS can improve the oxygen transmissibility of contact lenses, higher TRIS content will lead to lower equilibrium water content (EWC) and surface wettability of silicone hydrogel lens [8,9]. In addition, TRIS is prone to cause high lysozyme deposition on silicone-based hydrogel contact lenses [10]. Hydrophilic monomers including NVP, methacrylic acid (MAA), DMA, methacrylic acid (MA), methacrylic acid (MAA), HEMA, and glycerol methacrylate (GMA) were used to restrict the water absorbability decrease as well as poor wettability [11,12]. Additionally, NVP, HEMA, and DMA are hydrophilic monomers that have good water uptake ability and low cytotoxicity. For instance, HEMA can reduce albumin deposition on poly(HEMA-co-MMA) compounds [13]. Thus, this article is aiming to investigate the effect of these monomers on a silicone-based contact lens from TRIS, DMA, NVP and HEMA with respect to oxygen and water absorbability, protein uptake, and wettability. Accordingly, the resultant TRIS-DMA-NVP and TRIS-DMA-NVP-HEMA hydrogels were characterized by equilibrium water content, oxygen permeability, optical transparency, contact angle, mechanical properties, protein adsorption, and cytotoxicity.

## 2. Materials and Methods

### 2.1. Materials

N,N-dimethylacrylamide (DMA), 1-vinyl-2-pyrrolidinone (NVP), 1-hydroxycyclohexyl phenyl ketone (PI184), human serum albumin (HSA, MW 58 kDa), and lysozyme (MW 14.3 kDa) were purchased from Sigma-Aldrich (St. Louis, Mo USA). 3-(methacryloyloxy)propyltris(trimethylsiloxy) silane (TRIS, CAS No.: 17096-07-0) was bought from Alfa Aesar Chemical Co. (Heysham, Lancashire, UK). 2-hydroxyethylmethacrylate (HEMA) and ethylene glycol dimethacrylate (EGDMA) were obtained from Acros Organics (Morris Plains, NJ, USA). Phosphate buffered saline solution (PBS, 0.1 M, pH 7.4) was prepared in our laboratory.

### 2.2. Preparation of Silicone Hydrogels

The silicone hydrogels were polymerized from TRIS, DMA, NVP, and HEMA monomers in the presence of cross-linking agent EGDMA and photo initiator PI184, as listed in Table 1. The mixture was stirred in a dark environment at room temperature for 5 h. After being poured into polypropylene molds, the mixture was cured under 365 nm UV light for 40 min. After demolding, the lens material was purified by soaking in 50% ethanol for 24 h at 50 °C to remove the un-reacted monomers and photo initiator. Then, the lens was immersed in distilled water for 12 h at 50 °C to wash out the ethanol. Finally, the lens was preserved in PBS (pH 7.4) at room temperature. For all formulations, the percentages of EGDMA and PI184 were 0.625 wt % and 0.4 wt %, respectively.

### 2.3. Equilibrium Water Content

The equilibrium water content (EWC) of the sample was defined by the formula as follows: (1)$$\mathrm{EWC}\;(\%)=\frac{W_{2}-W_{1}}{W_{2}}\times 100$$
where W_1_ and W_2_ are the weights of the dry and wet lenses. The wet lens was rehydrated in distilled water at room temperature for 24 h.

### 2.4. Oxygen Permeability

The oxygen permeability (Dk, barrer) of the lens material was detected in humidified gas based on the polarographic method (ISO18369-4:2006) using an oxygen permeometer (Model 201T, Createch, Chesterfield Twp, MI, USA). The measurement was controlled at 35 ± 5 °C and 95% air humidity.

### 2.5. Optical Transparency

The lens was cut into a small piece (1 cm × 1 cm) after being fully hydrated in a PBS preserved solution. Then the sample was put into a cuvette containing 2 mL distilled water. The optical transparency of the sample was taken as the light transmittance (%) at 400–700 nm using a UV-Vis spectrophotometer (Cary 300, Agilient Technologies, Santa Clara, CA, USA).

### 2.6. Contact Angle [14]

The evaluation of the surface wettability of silicone hydrogel lens material was based on the contact angle test. The measurement was conducted using a contact angle goniometer (DSA 100, Krüss GmbH, Hamburg, Germany) at room temperature. The contact angle value was the average of five measurements.

### 2.7. Mechanical Properties

After swelling in distilled water, the specimens were cut into a dog bone shape. The dry specimens were obtained by putting them in an oven at 40 °C in one day. The tensile properties of the samples were measured at a crosshead speed of 50 mm/min, in accordance with ASTM D1708 standard, using a tensile tester (MTS 810, MTS Systems, Eden Prairie, MN, USA).

### 2.8. Protein Deposition [15]

The specimens (1 cm × 1 cm) were soaked in 1 mL PBS consisting of 2 mg/mL of lysozyme or HSA. The samples were washed three times with PBS (pH 7.4) after their placement in an incubator at room temperature for one day. Then, the samples were transferred into new tubes containing 2 mL of sodium dodecyl sulfate (SDS), and were shaken at 100 rpm for 1 h at room temperature. Afterwards, 2 mL bicinchoninic acid (BCA) was added and the tubes were put back in the incubator for 1 h at room temperature. Finally, protein deposition of all specimens was measured at 562 nm using a UV-Vis spectrophotometer (Cary 300, Agilient Technologies, Santa Clara, CA, USA).

### 2.9. Cytotoxicity [14]

Cytotoxicity of the hydrogel discs was determined according to ISO 10993-5. For determination of cytotoxicity, cell culture medium was prepared including 94% Dulbecco’s Modified Eagle’s medium (DMEM), 5% fetal bovine serum (FBS), and 1% penicillin antibiotic (PNC). The specimens (1 cm × 1 cm) were first placed under the UV light for 4 h to kill the bacteria. Then, the specimens were immersed in the cell culture medium at 37 °C for one day. After that, the extracted medium was passed through a 0.22 μL filter and collected in a new flask. After the cultures of L929 cells were completely initiated in the cell culture medium for more than one day at 37 °C, a controlled amount of L929 cells were transferred into the flask. The cell cultures were incubated at 37 °C for 48 h before thiazolyl blue tetrazolium bromide (MTT) reagent was added into the flask for 4 h at 37 °C. Then, dimethyl sulfoxide (DMSO) was mixed with the medium to dissolve the purple product. Finally, the absorbance at 570 nm was analyzed. In the cytotoxicity test, the positive control was simultaneously prepared and extracted with the negative control.

## 3. Results and Discussion

Two types of silicone hydrogel samples were prepared: TRIS-DMA-NVP and TRIS-DMA-NVP-HEMA. Table 1 lists the formulation of each type of silicone hydrogel.

### 3.1. Equilibrium Water Content

Equilibrium water content (EWC) is one of the important properties of the contact lens, offering more comfort for frequent patients, especially for the limitation of dry corneal eyes [16]. In TDN series, with equal content of DMA and NVP, EWC decreased sharply with the increase of TRIS content from 5 to 30 wt %. The highest EWC was 83.6%, corresponding to the minimum TRIS content of 5 wt %. The results demonstrate that water content of the lens material depended on the content of hydrophobic TRIS. In the TDNH series, with the incorporation of HEMA, EWC was also influenced by the TRIS content (from 20 to 40 wt %). A similar trend was reported in that the EWC of silicone-based hybrid hydrogels was affected by the ratio of NVP/DMA [17]. In particular, the increase of DMA ratio from 0 to 20 wt % led EWC to increase from 37% to 41%.

Multiple regression was performed on the EWC values in Table 1 and yielded the following correlation:

EWC(%) = 0.0784 C_T_ + 0.9197 C_D_ + 0.7800 C_N_ + 0.1131 C_H_  R^2^ = 0.9824
(2)
where C_T_, C_D_, C_N_ and C_H_ are the percentages of TRIS, DMA, NVP and HEMA in the lens, respectively. Based on this regression, TRIS contributed the least to EWC, due to its hydrophobic nature, whereas DMA contributed the most to EWC, due to its hydrophilicity. The log octane-water partition coefficient of a solute (logP) is a measure of the hydrophobicity. The values of logP for TRIS, DMA, NVP and HEMA are 7.36, −0.13, 0.37, and 0.47, respectively (given by US Environmental Protection Agency’s EPISuite™). Thus DMA is the most hydrophilic followed by NVP and HEMA, while TRIS is very hydrophobic. This trend agrees with the coefficients of the regressed correlation. It is possible to predict the EWC of the TRIS-DMA-NVP-HEMA system by using this regressed correlation.

### 3.2. Oxygen Permeability

Table 1 shows that the oxygen permeability (Dk) of silicone hydrogel lenses increased with the TRIS content from 5 to 30 wt %. The highest Dk value was 74.9 barrers obtained at a TRIS content of 40 wt %. The increase of Dk can be attributed to the incorporation of TRIS in a hydrogel network. Hydrophobic TRIS is a bulky polymer containing siloxane groups (-Si(CH_3_)_2_-O-) which manifests as a silicone domain in the hydrogel matrix. As such, the massive fibrous textures create small holes that oxygen molecules can pass easily to the corneal epithelium [1].

Figure 1 shows that the oxygen permeability decreased with the increase of EWC. Based on the linear correlation shown in Figure 1, we can predict the Dk of the silicone hydrogel from the EWC, which is in turn predictable from the regressed correlation in Section 3.1.

It has been reported that water content and oxygen transmissibility generally exhibit an inverse correlation. As stated in the literature, water is usually the limiting factor of oxygen transport [1]. In general, the silicon–oxygen bonds and free water are two major pathways for oxygen transportation in silicone hydrogel network. Because the oxygen absorptivity of siloxane group is 10 times greater than that of free water, oxygen transmissibility is primarily through the hydrophobic silicone phase, which is dominated by TRIS in this study. Other than the siloxane groups, oxygen can also transport through free water existing in the silicone phase [18]. In silicone a hydrogel matrix, water exists as free water and bound water. The bound water is directly connected to continuous hydrophilic phase consisting of non-freezable and freezable forms. On the other hand, the water trapped in hydrophobic silicone phase is mostly free water. As a result, in the silicone phase, small pores will appear with very low water content, and oxygen can be absorbed through this channel [19,20,21]. 

### 3.3. Optical Transparency

Light transmissibility is the measure of the transparency of contact lenses. Phase separation due to the unsuccessful cooperation of silicone and hydrogel groups would cause opaque lenses and restrict light transmissibility [22,23]. Figure 2 shows that the light transmittance of all hydrogel lenses varied around 96 to 97%, indicating that all of the samples are suitable for contact lenses, which requires the light transmittance to be above 90% [23].

### 3.4. Contact Angle

As an important factor influencing wearing comfort, the surface wettability of the lenses was evaluated by the contact angle of a drop of water on the solid surface. Figure 3 shows that the contact angle is linearly dependent on the EWC of the contact lens. Similar to predicting Dk, the linear correlation presented in Figure 3 can also be used to predict the contact angle from EWC.

Both contact angle and EWC are related to the hydrophilicity of the hydrogel. Hydrophobic TRIS decreased the hydrophilicity of the hydrogel, and hence decreased the wettability of lens surface. In addition, the hydrophobic monomer (TRIS) is sensitive to environmental conditions because of its reorientation property [24,25]. With the inherent flexible structure, silicon-oxygen bonds in TRIS monomers will rotate forth the lens surface in a dry environment [26]. Thus, TRIS is the monomer that influences the contact angle value of silicone hydrogel lens material according to a previous report [27].

As mentioned in Section 3.1, HEMA is less hydrophilic than DMA and NVP. At the same 30 wt % TRIS content, the contact angle of TDNH4 (14 wt % HEMA) was 75.4°, higher than TDNH3 (7 wt % HEMA), as well as TDN4 (0 wt % HEMA). Moreover, contact angle increased slightly with the decrease of NVP and increase of DMA for the same TRIS content. This means that NVP can increase the wettability more than DMA for silicone hydrogel polymers. In the literature, NVP was able to maintain good surface wettability during 4–6 h of wear, while DMA led to less wettable for the same wearing time [17]. In summary, in this work, almost all contact angles of hydrated lenses were less than 80°, which is comparable to that of other commercial contact lenses such as Lotrafilcon A (Air Optix Night & Day), and Balafilcon A (Purevision) [25].

### 3.5. Mechanical Properties

For contact lenses, low modulus offers the wearing comfort, physical impact, durableness, as well as fitting property [28], whereas high modulus can cause corneal swelling, papillary conjunctivitis, and corneal erosions due to corneal epithelial lesions [16].

Figure 4 shows that the modulus of hydrated lenses exhibited a linear dependence on the EWC, which is in agreement with previous studies [17,29]. Again, we can predict the modulus of this TRIS-DMA-NVP-HEMA silicone hydrogel from the composition through the regressed correlation of EWC.

The increase in modulus can be attributed to the hydrophobic characteristic of TRIS. As discussed in the Section 3.1, the existence of TRIS causes a decrease in the water content in hydrogel lens material. Lower EWC would then result in higher modulus and tensile strength. Moreover, because of the existence of free water in the hydrogel, as discussed in Section 3.2, these water molecules can move easily in and out of the hydrogel lens matrix and can evaporate faster than bound water. Thus, hydrogel lenses will become drier more easily and stronger in air without water or another preserving agent. In addition, the lower NVP content led to higher modulus and tensile strength, as reported in previous research [28]. Furthermore, the concentration and type of crosslinker may also influence the mechanical properties of silicone hydrogel-based contact lenses [30]. Overall, these silicone hydrogel polymers possess mechanical properties comparable to existing commercial contact lens materials.

### 3.6. Protein Adsorption

Protein is a fundamental factor that relates to the transport and metabolism of cellular membranes, potentially manifesting in antimicrobial or inflammation of the corneal eye [31]. Ophthalmic biomaterials for contact lenses should not adsorb constituents of tear film such as lipid and protein, because the deposition of protein can be uncomfortable, raise bacterial cell interaction to the corneal epithelium, and correlate with giant papillary conjunctivitis [32].

Basically, the adsorption of protein is related to the wettability of the lens. Figure 5 shows the effect of contact angle on the protein adsorption of all samples prepared in this work. The most hydrophilic lens (TDN1) exhibited the highest protein adsorption, while the most hydrophobic lens (TDNH6) exhibited the lowest protein adsorption. For HSA, when the TRIS content increased from 5 to 40 wt %, the adsorption of protein decreased from 5.31 to 2.77 µg/cm^2^ while lysozyme adsorption decreased from 8.85 to 3.42 µg/cm^2^. According to Figure 5, the deposition of lysozyme on the surface of hydrogel lens was higher than HSA absorption. This phenomenon is due to the lower molecular weight of lysozyme.

The protein deposition of contact lens is also influenced by other factors such as pH of environment, protein dimension, charge of protein, lens material, water content, roughness of lens surface, and material properties [33,34]. In previous research, protein adsorption was directly proportional to the increase of hydrophobic materials as well as TRIS content, which resulted in decreasing water content [35,36]. However, yet other research indicates that increasing protein adsorption was influenced by loading hydrophilic components, especially NVP [32,37,38]. The lysozyme deposition of five silicones-based hydrogel lens material was less than that of pHEMA and decreased with decreasing water content [39]. This may be attributed to the property of the lactam moiety in the NVP [38]. Lysozyme is positively charged whereas HSA is negatively charged [13]. In the molecule structure of NVP, the negative charge belongs to carbonyl group and lysozyme can be attracted to this group. Similarly, HSA will be attracted by the positive charge of nitrogen end. In conclusion, the decrease of protein deposition of all formulations was mainly due to the NVP content in silicone hydrogel.

### 3.7. Cytotoxicity

On ophthalmic devices, cellular behavior is significant determining the biocompatibility of cells [9]. A cytotoxicity test was conducted to observe the change of cellular morphology by adding the extracted medium of hydrogel lens to a cell culture. Figure 6 shows all extracted media of lenses did not clearly affect cellular growth. The cell mass for all hydrogel lenses increased and was comparable to the negative control. Therefore, the result demonstrates that the hydrogel composing of TRIS, DMA, NVP and HEMA was non-cytotoxic. The RGR value changed continuously from day 1 to day 3 for all extracted medium. In summary, this result proved that the combination of TRIS-DMA-NVP and TRIS-DMA-NVP-HEMA were safe for cells and humans.

### 3.8. Comaprison with Commercial Lenses

Table 2 summarizes the properties of commercial lenses and the lenses in this study. Commercial lenses such as Air Optix, Acuvue Oasys, PureVision exhibited higher Dk and lower EWC than the lenses in this study. Although the TDNH6 lens exhibited a maximum contact angle of 81.9° and Young’s modulus of 1.29 MPa, it was still lower than that of PureVision and Air Optix Night & Day, respectively. In general, our hydrogel lenses in this work had EWC, Dk, contact angle and Young’s modulus comparable to commercial lenses. However, our lenses still exhibited lower Dk than PureVision, Acuvue Oasis and Air Optix Night & Day. Thus this formulation still requires further improvement.

## 4. Conclusions

As demonstrated in this study, the silicone hydrogels were polymerized from TRIS, DMA, NVP, and HEMA monomers. By changing the compositions, the hydrophilicity of the resultant silicone hydrogel lenses can be adjusted. An empirical correlation was obtained through multiple regression for the equilibrium water content. The EWC increased with the contents of DMA and NVP while decreasing with the increase TRIS. Furthermore, the oxygen permeability, the contact angle and Young’s modulus exhibited linear dependency on EWC. With increasing EWC, the Dk, the contact angle, and the modulus all decreased. The deposition of human serum albumin and lysozyme decreased with the contact angle, and the proliferation of L929 indicated that these lenses were non-cytotoxic. Overall, all samples presented high light transparency, reasonable moduli, and non-cytotoxicity. The results of this research showed that these hydrogels exhibited EWC, Dk, contact angle and Young’s modulus comparable with commercial lenses such as Biomedics, Acuvue Advance, Acuvue Oasys, and PureVision. Therefore, the results of this work would be useful for developing soft contact lenses.

## Figures and Tables

**Figure 1 polymers-11-00944-f001:**
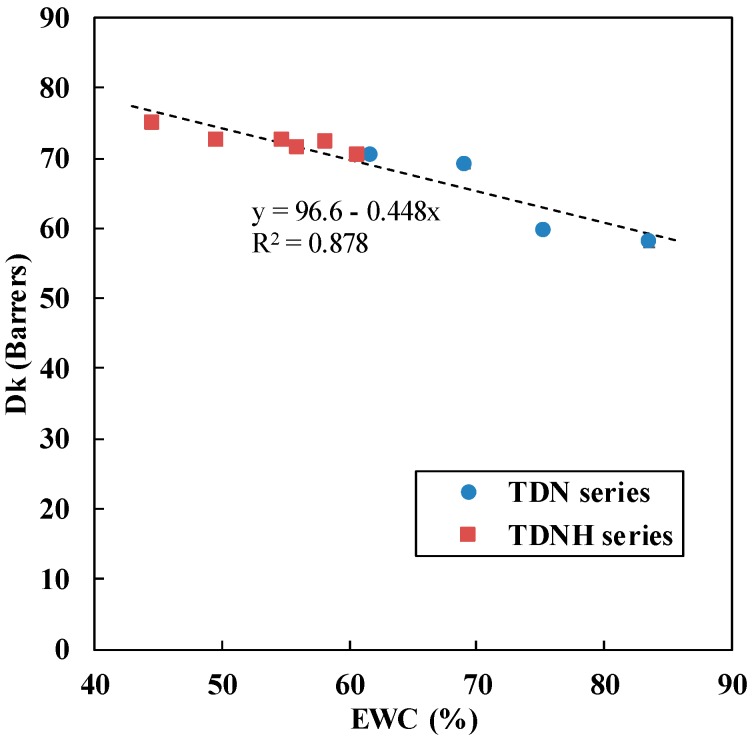
The correlation between Dk and EWC.

**Figure 2 polymers-11-00944-f002:**
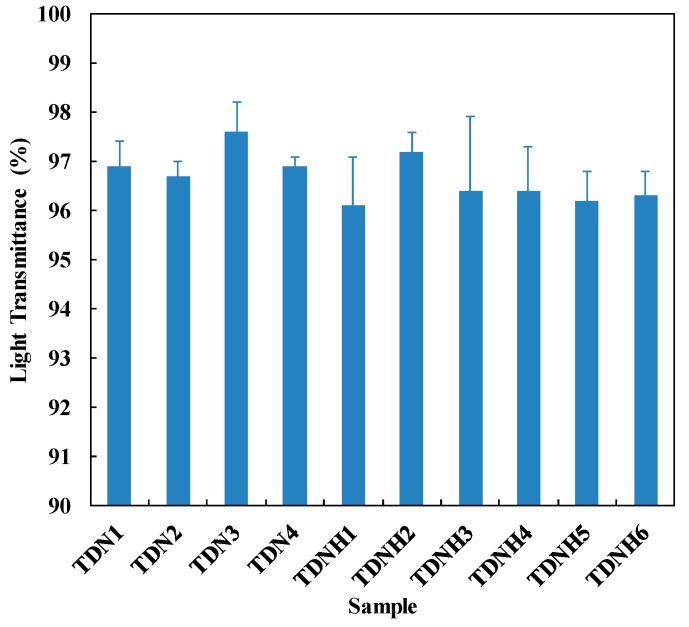
Light transmittance of hydrated modified silicone hydrogel samples.

**Figure 3 polymers-11-00944-f003:**
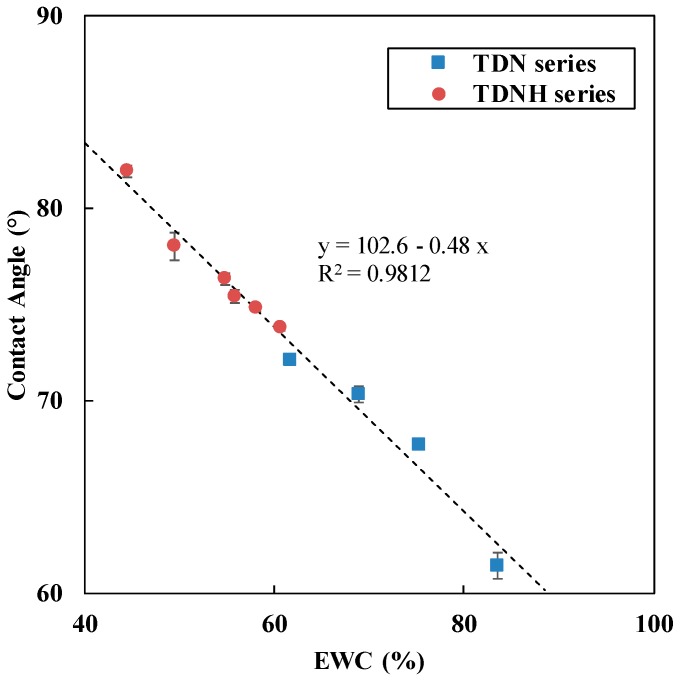
The correlation between EWC and contact angle for p(TRIS-DMA-NVP-HEMA).

**Figure 4 polymers-11-00944-f004:**
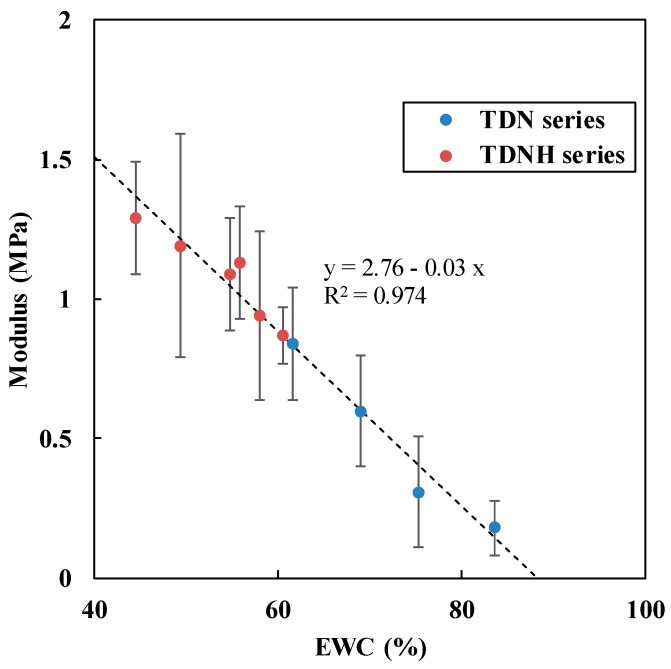
The correlation between the modulus and EWC for p(TRIS-DMA-NVP-HEMA).

**Figure 5 polymers-11-00944-f005:**
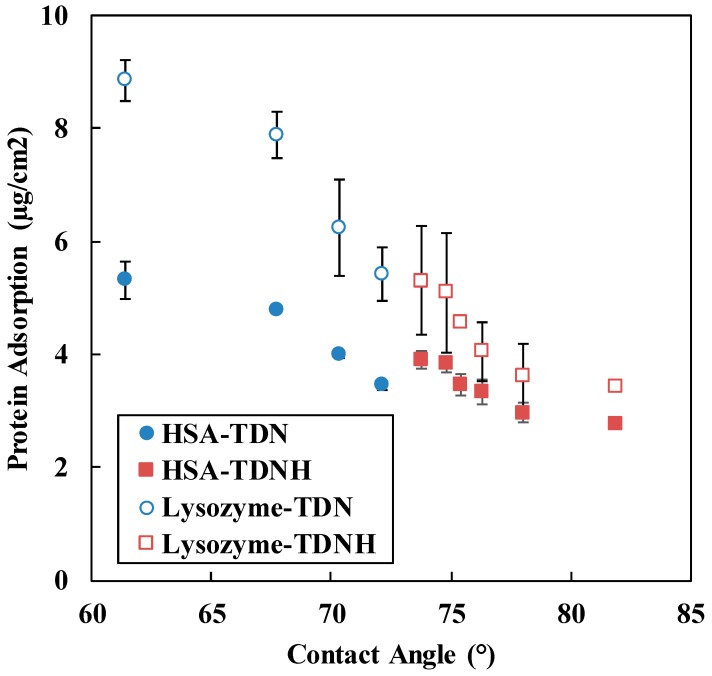
The effect of contact angle on the adsorption of lysozyme and HAS.

**Figure 6 polymers-11-00944-f006:**
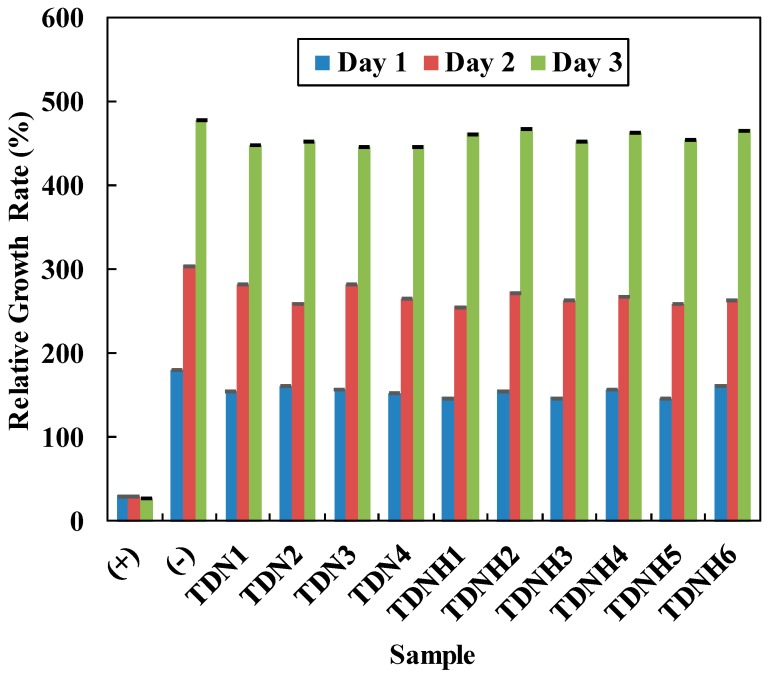
Relative growth rate (RGR) of silicone hydrogel specimens during 3 cultured days with L929 cells of lens samples.

**Table 1 polymers-11-00944-t001:** Formulations of silicone hydrogel by copolymerization of TRIS, DMA, NVP.

Sample	Feed (wt %)				EWC (%)	Dk (barrer)	Contact Angle (°)	Modulus (MPa)
	TRIS	DMA	NVP	HEMA				
TDN1	5	47.5	47.5	0	83.6 ± 0.4	58.0	61.4 ± 0.7	0.56 ± 0.20
TDN2	10	45	45	0	75.3 ± 1.5	59.8	67.7 ± 0.2	0.65 ± 0.12
TDN3	20	40	40	0	69.0 ± 0.7	69.0	70.3 ± 0.4	0.70 ± 0.21
TDN4	30	35	35	0	61.7 ± 1.8	70.4	72.1 ± 0.3	0.84 ± 0.15
TDNH1	20	24	48	8	60.6 ± 0.7	70.6	73.8 ± 0.2	0.87 ± 0.10
TDNH2	20	40	24	16	58.1 ± 0.3	72.3	74.8 ± 0.2	0.94 ± 0.29
TDNH3	30	21	42	7	55.9 ± 0.1	71.4	75.4 ± 0.3	1.13 ± 0.18
TDNH4	30	35	21	14	54.8 ± 0.9	72.6	76.3 ± 0.3	1.09 ± 0.18
TDNH5	40	18	36	6	49.5 ± 1.0	72.5	78.0 ± 0.7	1.19 ± 0.39
TDNH6	40	30	18	12	44.5 ± 1.1	74.9	81.9 ± 0.3	1.29 ± 0.19

**Table 2 polymers-11-00944-t002:** Properties comparison of commercial lenses and this study.

Product	Manufacturer	Dk (barrers)	EWC (%)	Contact Angle (°)	Modulus (MPa)	Principle Monomers
Biomedics 38	CooperVision	8.4	38	30	0.81	HEMA, EGDMA
Acuvue 2	Johnson & Johnson Vision Care	19	58			HEMA, MAA, EGDMA
Biomedics XC	CooperVision	44	60			HEMA, MAA, PC, TEGDMA
Acuvue Advance	Johnson & Johnson	60	47	65.6	0.43	MPDMS, DMA, HEMA, EGDMA, siloxane macromer, PVP
PureVision	Bausch & Lomb	91	36	93.6	1.10	TEGDMA, NVP, TPVC, NCVE, PBVC
Acuvue Oasys	Johnson & Johnson	103	38	78.7	0.72	MPDMS, DMA, HEMA, siloxane macromer, TEGDMA, PVP
Air Optix	CIBA Vision	110	33	44.4	1.00	DMA, TRIS, siloxane monomer
Air Optix Night & Day	CIBA Vision	140	24		1.52	DMA, TRIS, siloxane monomer
TDN3	This work	69.0	69.0	70.3	0.70	TRIS, DMA, NVP
TDN4		70.4	61.7	72.1	0.84	
TDNH4		72.6	54.8	76.3	1.09	TRIS, DMA, NVP, HEMA
TDNH6		74.9	44.5	81.9	1.29	

^PVP^ polyvinyl pyrrolidone; ^MPDMS^ monofunctional polydimethylsiloxane; ^DMA^ N,M-dimethylacrylamide; ^EGDMA^ ethyleneglycol dimethacrylate; ^HEMA^ hydroxyethyl methacrylate; ^TEGDMA^ tetraethyleneglycol dimethacrylate; ^TRIS^ trimethyl siloxysilyl; ^NVP^ N-vinyl pyrrolidone; ^TPVC^ tris-(trimethyl siloxysilyl)propylvinyl carbamate; ^NCVE^ N-carboxyvinyl ester; ^PBVC^ poly-(dimethysiloxy) di-(silylbutanol) bis-(vinyl carbamate).

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
