# Peer review of "Synthesis and Characterization of Silicone Contact Lenses Based on TRIS-DMA-NVP-HEMA Hydrogels"

_polymers, 2019, doi:10.3390/polym11060944_

Reviewer 1 Report

The article

Synthesis and characterization of silicone contact 2 lenses based on TRIS-DMA-NVP-HEMA hydrogelsby

 Nguyen-Phuong-Dung Tran and Ming-Chien Yang

deals with interesting approach the studies of the study, silicone-based hydrogel contact lenses were prepared by the polymerization of 3-(methacryloyloxy)propyltris(trimethylsiloxy)silane (TRIS), N,N-dimethylacrylamide (DMA),  1-vinyl-2-pyrrolidinone (NVP), and 2-hydroxye-thylmethacrylate (HEMA). The properties of  silicone hydrogel lenses were analyzed based on the methods such as equilibrium water content, oxygen permeability, optical transparency, contact angle, mechanical test, protein adsorption, and cell toxicity. As a result of the measurements carried out, the authors showed that an empirical correlation was obtained through multiple regression for the equilibrium water content. The EWC increased with the contents of DMA and NVP while decreased with the increase TRIS. Furthermore, the oxygen permeability, the contact angle and  Young’s modulus exhibited linear dependency on EWC. With increasing EWC, the Dk, the contact angle and the modulus all decreased.  In the end of conclussion the authors  state that this work may be useful for ophthalmic contact lenses.

The experiments conducted by the authors, calculations, figures, results, discussion and conclusion are very interesting and correct. So this article surely can be published in Polymers

But, some minor revisions could be suggested to authors to improve soundness of the paper.

Please fill in the following issues:

1. Taking into account the results of experiments and their interpretations, the authors presented too modest conclusions in the "Conclusion". I propose to expand more boldly applications, especially in the direction of application of contact lenses.

Author Response

Thank you for your appreciation in our work. As per your suggestion, we have re-written the conclusion to include our expectation for this paper.

The results of this research showed that these hydrogels exhibited EWC, Dk, contact angle and Young’s modulus comparable with commercial lenses such as Biomedics, Acuvue Advance, Acuvue Oasys, and PureVision. Therefore, the results of this work would be useful for developing soft contact lenses.

Reviewer 2 Report

The manuscript investigated equilibrium water content, oxygen permeability, optical transparency, contact angle, mechanical test, protein adsorption, and cell toxicity of silicone-based contact lens containing TRIS, DMA, NVP and HEMA. Generally, experiments are well carried out with results that seems to be interests to many of authors. There are few concerns that may need consideration before the acceptance of this article.

It is unclear how the test groups were decided. There seems to be variable of TRIS, DMA, NVP contents without NVP for TDN groups, while variable concentration of all 4 components in TDNH. How the numbers (wt%) were decided? Is there any preliminary studies?

What is the control in this study? Is material without HEMA is the control? Also, how these material would be compared to the commercial products? Is there need of including commercial products as control?

For contact angle, how the method has been decided? What liquid has been used (Water? Glycerol?). What is the reason for choosing the liquid? When did you measure the contact angle (following how many seconds or minutes after the drop?)

Protein deposition test seems to be lacking information. Where does protein came from? (Is there a step where immersion into protein solution exist?) Why the SDS is used which is surfactant?

Cytotoxicity test indicate non-toxicity of the material as stated in ISO 10993-5. It may not be appropriate to indicate it as cell proliferation. Also, L929 cells are fibroblasts and it would be ideal to indicate relevance of cell lines with respect to corneal cells.

Author Response

1. It is unclear how the test groups were decided. There seems to be variable of TRIS, DMA, NVP contents without NVP for TDN groups, while variable concentration of all 4 components in TDNH. How the numbers (wt%) were decided? Is there any preliminary studies?

Reply:

    Our original formulation was suggested by a local contact lens manufacturer. They used TRIS, DMA and NVP. The original formulation was modified to improve the stiffness and Dk.

2. What is the control in this study? Is material without HEMA is the control? Also, how these material would be compared to the commercial products? Is there need of including commercial products as control?

Reply:

    We didn't assign the control in this study. However, as per your suggestion, we added 4 commercial lenses in the revision for comparison.

3. For contact angle, how the method has been decided? What liquid has been used (Water? Glycerol?). What is the reason for choosing the liquid? When did you measure the contact angle (following how many seconds or minutes after the drop?)

Reply:

    The contact angle was taken within 5 sec after placing a drop of water. The measurement was adopted from the following papers:

[1]  Abbasi, F; Mirzadeh, H; Simjoo, M. Hydrophilic interpenetrating polymer networks of poly (dimethyl siloxane)(PDMS) as biomaterial for cochlear implants. J Biomater Sci Polym Ed. 2006;17(3):341-355.

[2] Van Beek, M; Weeks, A; Jones, L; Sheardown, H. Immobilized hyaluronic acid containing model silicone hydrogels reduce protein adsorption. J Biomater Sci Polym Ed. 2008;19(11):1425-1436.

[3]  Korogiannaki, M; Guidi, G; Jones, L; Sheardown, H. Timolol maleate release from hyaluronic acid-containing model silicone hydrogel contact lens materials. J Biomater Appl. 2015;30(3):361-376.

4. Protein deposition test seems to be lacking information. Where does protein came from? (Is there a step where immersion into protein solution exist?) Why the SDS is used which is surfactant?

Reply:

    The method for determining protein deposition was adopted from the following papers:

[1]  Ishihara, K; Fukumoto, K; Iwasaki, Y; Nakabayashi, N. Modification of polysulfone with phospholipid polymer for improvement of the blood compatibility. Part 1. Surface characterization. Biomaterials. 1999;20(17):1545-1551.

[2]  Inoue, Y; Watanabe, J; Ishihara, K. (2004). Dynamic motion of phosphorylcholine groups at the surface of poly (2-methacryloyloxyethyl phosphorylcholine–random–2, 2, 2-trifluoroethyl methacrylate). J Colloid Interface Sci. 2004;274(2):465-471.

    The proteins (HSA and lysozyme) were purchased from Sigma-Aldrich, USA. The sample was immersed in the PBS solution containing the protein for 1 day. SDS was used to desorb deposited proteins from the surface.

5. Cytotoxicity test indicate non-toxicity of the material as stated in ISO 10993-5. It may not be appropriate to indicate it as cell proliferation. Also, L929 cells are fibroblasts and it would be ideal to indicate relevance of cell lines with respect to corneal cells.

Reply:

    The cytotoxicity was determined based on the paper below:

[1] Abbasi, F; Mirzadeh, H; Simjoo M. Hydrophilic interpenetrating polymer networks of poly (dimethyl siloxane)(PDMS) as biomaterial for cochlear implants. J Biomater Sci Polym Ed. 2006;17(3):341-355.

    ISO 10993-5 is the standard for in-vitro testing cytotoxicity of biomaterials. L929 is most often used for ISO 10993-5.

Round  2

Reviewer 1 Report

The corrections introduced by the authors are satisfactory.

Reviewer 2 Report

All comments are well addressed.